# An iris diaphragm mechanism to gate a cyclic nucleotide-gated ion channel

Arin Marchesi[1,2], Xiaolong Gao[3], Ricardo Adaixo[4], Jan Rheinberger[3], Henning Stahlberg [4], Crina Nimigean[3,5,6] & Simon Scheuring [1,3,5]

Cyclic nucleotide-gated (CNG) ion channels are non-selective cation channels key to signal transduction. The free energy difference of cyclic-nucleotide (cAMP/cGMP) binding/unbinding is translated into mechanical work to modulate the open/closed probability of the pore, i.e., gating. Despite the recent advances in structural determination of CNG channels, the conformational changes associated with gating remain unknown. Here we examine the conformational dynamics of a prokaryotic homolog of CNG channels, SthK, using high-speed atomic force microscopy (HS-AFM). HS-AFM of SthK in lipid bilayers shows that the CNBDs undergo dramatic conformational changes during the interconversion between the resting (apo and cGMP) and the activated (cAMP) states: the CNBDs approach the membrane and splay away from the 4-fold channel axis accompanied by a clockwise rotation with respect to the pore domain. We propose that these movements may be converted by the C-linker to pull the pore helices open in an iris diaphragm-like mechanism.

[1] INSERM U1006, Aix-Marseille Université, Parc Scientifique et Technologique de Luminy, 163 Avenue de Luminy, 13009 Marseille, France. [2] INSERM U1067, Aix-Marseille Université, Parc Scientifique et Technologique de Luminy, 163 Avenue de Luminy, 13009 Marseille, France. [3] Department of Anesthesiology, Weill Cornell Medical College, 1300 York Ave, New York, NY 10065, USA. [4] Center for Cellular Imaging and NanoAnalytics, Biozentrum, University of Basel, Mattenstrasse 26, 4058 Basel, Switzerland. [5] Department of Physiology and Biophysics, Weill Cornell Medical College, 1300 York Ave, New York, NY 10065, USA. [6] Department of Biochemistry, Weill Cornell Medical College, 1300 York Ave, New York, NY 10065, USA. Correspondence and requests for materials should be addressed to C.N. (email: crn2002@med.cornell.edu) or to S.S. (email: sis2019@med.cornell.edu)

Cyclic nucleotide-gated (CNG) channels and hyperpolarization-activated cyclic nucleotide-regulated (HCN) channels constitute a key physiological link between the cyclic nucleotide (cN) intracellular second messenger system and electrical signaling at the plasma membrane[1–3]. These channels are cation-permeable and are regulated by the direct binding of cNs (cAMP or cGMP) to a specialized intracellular domain, the cyclic nucleotide-binding domain (CNBD), which is also found in other signaling proteins like the bacterial transcription factor CAP and protein kinases (PKA and PKG)[4,5]. The free energy difference of CNs binding/unbinding is transduced into a mechanical conformational change opening/closing the channel pore, which in turn leads to an electrical response by allowing/prohibiting ion flow across the cell membrane.

Genetic and pharmacological studies highlight the importance of these channels in numerous processes ranging from neuronal excitability in the brain and pacemaking in the heart, to sensory transduction in the retina and olfactory epithelium[6–9]. Thus, elucidating how these channels respond to endogenous ligands or drugs has great physiological and clinical significance.

CNG and HCN channels belong to the voltage-gated ion channel superfamily and—as all members—form tetramers whose subunits surround a central pore[10]. Each monomer consists of six transmembrane (TM) helices (S1–S6); the first four (S1–S4) form the voltage-sensor domain (VSD) and the last two helices (S5–S6) of the four subunits assemble to form the pore domain (PD) (Fig. 1a). The CNBD is C-terminal and connected to S6 via the C-linker (Fig. 1a), a highly conserved domain within the family, believed to provide the functional coupling between ligand-binding and channel opening[11,12]. The CNG and HCN channels families (Fig. 1b) have been extensively studied with electrophysiology, and, despite their significant homology in sequence and architecture, they differ greatly in their gating modalities. While HCN channels are gated by hyperpolarization and only modulated by cN-binding, CNG channels display little voltage dependence and are gated by CN-binding to the CNBD[2,13,14]. Structural approaches have only recently become successful with the advent of high-resolution cryo electron microscopy (cryo-EM)[15].

The cryo-EM structures of the closed-pore apo and cAMP-bound human HCN1 channel, the open-pore structure of the cGMP-bound CNG channel TAX4 from *Caenorhabditis elegans*, and the structure of a prokaryotic CNG channel homolog LliK, represented a breakthrough in the field[16–18]. However, the TAX4 and the LliK channels are only in cAMP-bound states, and a comparison between the two very similar HCN1 apo and

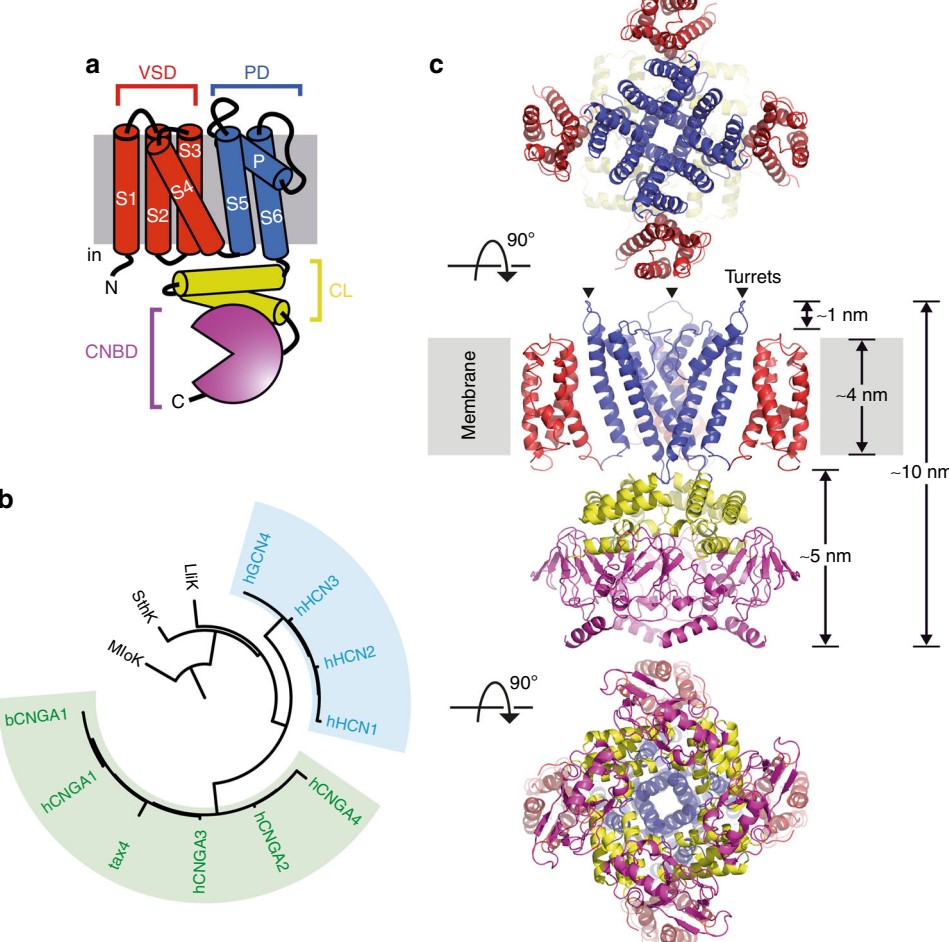

**Fig. 1** SthK is a cyclic nucleotide-gated ion channel related to the CNG and the HCN subfamilies. **a** Cartoon of the topology of one subunit. The different functional domains are highlighted in colors: voltage sensor domain (VSD, red), pore domain (PD, blue), C-linker (CL, yellow), and cyclic nucleotide binding domain (CNBD, purple). **b** Phylogenetic tree of the CNGA (green) and HCN (blue) subfamilies. The prokaryotic CNG channels LliK and MloK1 (lacking the CL), of which the structures are known, are shown for comparison. **c** High-resolution cryo-EM structure of the SthK resting state (PDB 6CJQ) viewed from the extracellular (top), membrane plane (middle), and intracellular (bottom) side. The functional domains are color-coded as in **a**. Turrets are highlighted by arrowheads. For clarity, the VSD and PD domains of the protomer nearer to the viewer have been omitted in the membrane plane view

cAMP-bound structures suggests that the conformational changes upon ligand-binding are minor in this channel[16]. This is consistent with the little functional effect seen in HCN1 channels electrophysiology studies upon cAMP addition[19]. Therefore, despite the recent structural progress, there is a need for a direct assessment of conformational changes in a functional CNG channel in physiologically relevant conditions.

Here we use SthK, an excellent functional and structural model of CNG channels (Fig. 1b, c), to investigate the conformational changes and dynamics associated with cN-binding using high-speed atomic force microscopy (HS-AFM) in real time[20–23]. Three structures of SthK have now been solved, in apo, cGMP-bound and cAMP-bound states[22]. All these structures are in the resting state, which is consistent with the low open probability upon cAMP-binding, however, the absence of an open state highlights the difficulty in capturing infrequent states with cryo-EM[20–22]. On the other hand, HS-AFM, which can monitor the same molecule under different conditions in real time, is ideal for investigating this process. HS-AFM allows the observation of single molecule surfaces with high lateral (~1 nm), vertical (~0.1 nm), and temporal (~100 ms) resolution under physiological conditions, i.e., in membrane, immersed in buffer solution, at ambient temperature and pressure, without labeling, staining or fixing procedures[24–29]. Here, we show that channel activation in response to cAMP-binding is associated with large, concerted and reversible conformational changes. The rearrangement from a resting to an activated state includes a vertical CNBD movement towards the membrane, as well as splaying and rotation of the CNBDs. We provide also indirect evidence that these movements in the CNBDs are accompanied by rearrangements in the VSDs. Single channel recordings show that the conformational changes we see with HS-AFM correspond to drastic changes in channel gating, allowing straightforward correlation between channel conformations and functional states.

## Results

**HS-AFM imaging of SthK in the cAMP-bound state.** SthK was reconstituted in bilayer membranes at low lipid-to-protein ratio in the presence of saturating, 0.1 mM cAMP[20,21]. For HS-AFM analysis, these membranes were adsorbed onto freshly cleaved mica and imaged in physiological buffer (see Methods). We identified large membrane patches (200–300 nm in diameter) with densely packed SthK channels (Supplementary Fig. 1a, inset 1), and smaller patches (50–100 nm) with well-ordered SthK 2D-crystals (Supplementary Fig. 1a, inset 2). The densely packed areas revealed a strongly corrugated surface profile (Supplementary Fig. 1b, top), originating from the alternating up-and-down packing of SthK channels with respect to the membrane plane. Taking advantage of the up-and-down packing, we measured a height difference of 4.6 nm for the cytoplasmic domains (C-linker and CNBD) above the membrane (Supplementary Fig. 1c), in good agreement with the SthK cryo-EM structure (Fig. 1c)[22]. For high-resolution imaging we focused on the ordered 2D-crystals (Supplementary Fig. 1b, bottom) because they provide an ideal platform for high-resolution structural analysis due to their regular packing (Fig. 2), and only molecules in this packing responded to changes in cyclic nucleotides, as shown below (Figs. 3, 4). These small 2D-crystals represented approximately 10% of the observed sample.

The SthK 2D-crystals in 0.1 mM cAMP were imaged with HS-AFM from both extracellular (Fig. 2a–d) and intracellular (Fig. 2e–g) sides. The large differences between the membrane protrusion heights on the extracellular and the intracellular sides (Supplementary Fig. 1a,b) allowed unambiguous assignment of the channel orientations and their bilayer packing in these 2D-crystals, where all molecules are inserted in the same orientation.

Topographs where the heights above the membrane were low (~0.6 nm) were deemed to represent the channels viewed from the extracellular face. From this side, we see the tetrameric arrangement of the turrets, the loops between the S5 and pore helix that protrude the furthest into the extracellular space (Figs. 2a, 1c, middle panel). Interestingly, despite the unidirectional insertion of the channels in the bilayer, analysis of high-resolution images (Fig. 2a, b) showed that neighboring SthK molecules in the unit cell are rotated and vertically shifted in the membrane plane with respect to each other. Cross-section analysis indicated that the higher molecules emerged $0.6 \pm 0.1$ nm and the lower $0.3 \pm 0.1$ nm from the membrane (Fig. 2c, left). The small height difference of ~0.3 nm (Fig. 2c, right, Supplementary Fig. 2a) corroborates that the molecules within the lattice are unidirectionally inserted (as the intracellular side of the channel would protrude several nanometers from the membrane, see Supplementary Fig. 1c and Fig. 1c, middle). In agreement with this interpretation, no morphological difference is observed in the correlation-average topography between the lower and upper tetramer (Fig. 2b). This rather rare packing pattern in a 2D-crystal (Supplementary Fig. 1b, bottom panel) has previously been observed in AQP2 2D-crystals[30]. An atomic model of the SthK channel viewed from the extracellular face can precisely be docked into the protein lattice (Fig. 2d), without clashes of the peripheral VSDs (the fainter densities in Fig. 2d), which do not protrude from the membrane and are therefore not visible in the topograph (compare Fig. 2b, d). Given that the turrets on the extracellular face of the PD are small protrusions, the HS-AFM is able to contour them individually, thus allowing us to define the orientation of the PD with high precision.

Viewed from the opposite side, deemed to be the cytoplasmic face due to the large protrusions (~4.6 nm) originating from the channel C-linkers and CNBDs, the packing appears less obvious (Fig. 2e). The average unit cell (Fig. 2f) has the global appearance of a large tetragonal star that is nevertheless too large to represent just one channel tetramer. However, upon close inspection and careful height analysis, the arrangement of molecules becomes visible, where adjacent molecules are packed with different rotation and height levels (Fig. 2e–g). The unit cell dimensions, section profiles, and height differences between neighboring molecules reveal the same packing as observed from the extracellular face (Fig. 2f, g, Supplementary Fig. 2b). The height analysis showed the same ~0.3 nm height difference between neighboring channels, although on this face the channels protrude $4.3 \pm 0.1$ and $4.6 \pm 0.1$ nm from the membrane, respectively (Fig. 2g), in agreement with the expected height of the C-linker/CNBD (Fig. 1c).

Interestingly, the 2D-crystal packing model of the resting-state SthK channel structure that we built in order to fit the extracellular view shown in Fig. 2b no longer fits the topography of the intracellular view (compare Fig. 2f with Fig. 2h). This indicates that the CNBDs assume a different configuration than that seen in the resting SthK channel structure because the orientation of the extracellular view topography can be unambiguously assigned to the very well-defined structural turrets (Fig. 1c). Indeed, a clockwise (CW) rotation by 25° of the CNBDs relative to the PD is needed to match the HS-AFM topography (Fig. 2i, compare to f). Thus, during activation, the CNBD undergoes a 25° CW rotation compared to the resting state.

**Real-time HS-AFM imaging of SthK conformational changes.** In order to image conformational changes upon channel closing, we needed to either wash the cAMP away, in order to obtain the apo closed conformation, or replace the cAMP with cGMP.

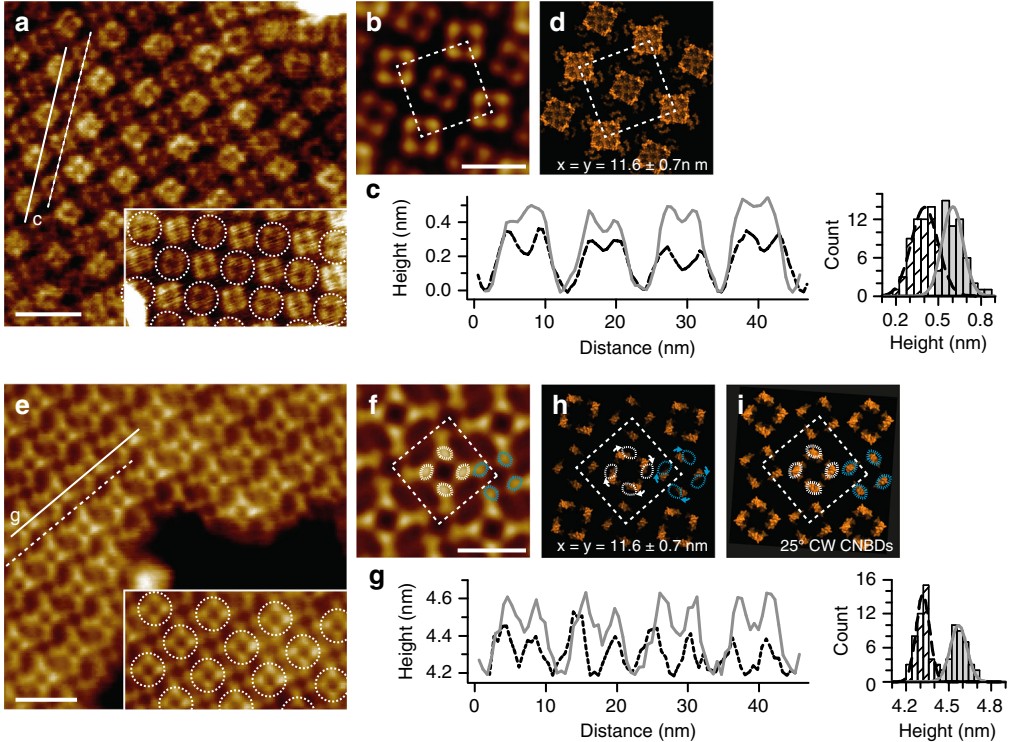

**Fig. 2** Characterization of the SthK 2D-crystals in presence of cAMP. **a** HS-AFM image of a 2D-crystal viewed from the extracellular side. Each channel appears as a square-shaped tetramer. Each protruding turret of the pore domain within the channel tetramer is clearly resolved. Inset: The alternating packing of the channels is highlighted with dashed outlines (scale bar: 15 nm). **b** Four-fold symmetrized correlation average of **a**. The unit cell (dashed square, dimensions $a = b = 11.6$ nm, $\gamma = 90°$) comprises two tetramers (scale bar: 10 nm). **c** Left: Height profiles of SthK molecules along the dashed and solid lines in **a** are shown by the black and grey traces, respectively. Right: Height histogram of the two classes of molecules with 0.3 and 0.6 nm average protrusion heights. **d** Packing model of the 2D-crystal (PDB 6CJQ). **e**, **f**, **g**, **h** same as **a**, **b**, **c**, and **d**, but the 2D-crystal is imaged from the intracellular side exposing the CNBDs. Dashed circles in **a** and **e** highlight the alternating packing of the tetramers in the 2D-crystal. The stronger-protruding CNBDs on the intracellular side (**e**, **f**) are the less protruding ones from the extracellular side (**a**, **b**). White arrows in **h** indicate the CW rotation that the CNBDs should undergo upon cAMP-binding to match the HS-AFM data. The dotted white and cyan circles in **h** show the position of the protrusions in the data, which do not match the packing of the resting state SthK model. **i** Structural model with 25° CW rotated CNBDs matches the experimental data (compare with **h**). The white and cyan dashed circles in **f** highlight the two neighboring tetramers of high and low height, respectively. The 25° CW rotation of the CNBDs in the model brings the model CNBD back into the white and cyan dashed circles (**i**)

Previously, it was shown that, unlike for other CNG channels, cGMP is not an agonist for SthK channels but inhibits cAMP-induced activity and thus also leading to channel closing[20,21]. We imaged using HS-AFM the transition to the resting/closed state of SthK from an activated/open state by either washing the cAMP away from a 2D-crystal previously imaged in 0.1 mM cAMP (Supplementary Fig. 3) or by adding excess cGMP. Although both experiments led to the same qualitative changes in the 2D-crystals (Fig. 3a, compare with Supplementary Fig. 3), we focused our analysis on the cGMP transition because there is always uncertainty regarding the timing and the completion of the ligand removal from the binding pocket upon simply washing the ligand away[31,32]. After repeatedly imaging the intracellular face of a cAMP-bound SthK 2D-crystal for ~5 min without observing any structural changes, we added cGMP to a final concentration of 7 mM (Supplementary Movie 1; Fig. 3a). A marked change in the surface topography and 2D-crystal packing was observed, which initiated from the molecules located at the crystal patch border and propagated with slow kinetics toward the center (dashed outline in Fig. 3c). Note, despite the fact that the transition in the 2D-crystal is significantly slowed down (~150 s), and each image is taken at 1 s⁻¹, each molecule in these frames (scanned by ~15 scan lines, where the full image contains 300 scan lines and thus the line acquisition speed is 3.3 ms) existed for ~50 ms in these images. Once the transition was completed, no further

changes were observed for the remainder of the recording. Given the dramatic morphological changes observed, we wanted to correlate this transition to functional phenotypes. For this, we reconstituted the same purified SthK channels used to grow the 2D-crystals for HS-AFM, into liposomes for electrophysiological experiments. SthK channels incorporated into planar lipid bilayers showed large single-channel currents of ~4 pA single-channel amplitude at 0.1 mM cAMP (~20-fold higher concentration than the reported apparent cAMP affinity[20,21]). The channel activity was largely inhibited in the presence of milli-molar cGMP[20,21]: the open probability ($P_O$) decreased 1000-fold from ~0.25 to ~0.0003 at +100 mV (Fig. 3b, top and middle traces). Both the channel activity and number of active channels in the bilayer were fully recovered upon re-addition of cAMP (Fig. 3b, bottom trace).

It is interesting to note that since the channel activity is voltage-dependent, the open probability becomes even lower when the voltage is 0 mV[21], the membrane voltage in the AFM and cryo-EM experiments. Thus, one would expect that most of the SthK molecules in the membrane patches are in resting rather than active conformations. And indeed, the densely packed membranes shown in Supplementary Fig. 1a, inset 1, contain channels that do not change conformation upon cAMP removal, suggesting that they are in the resting state. We speculate that the reason we have 2D crystals of SthK in an active conformation is

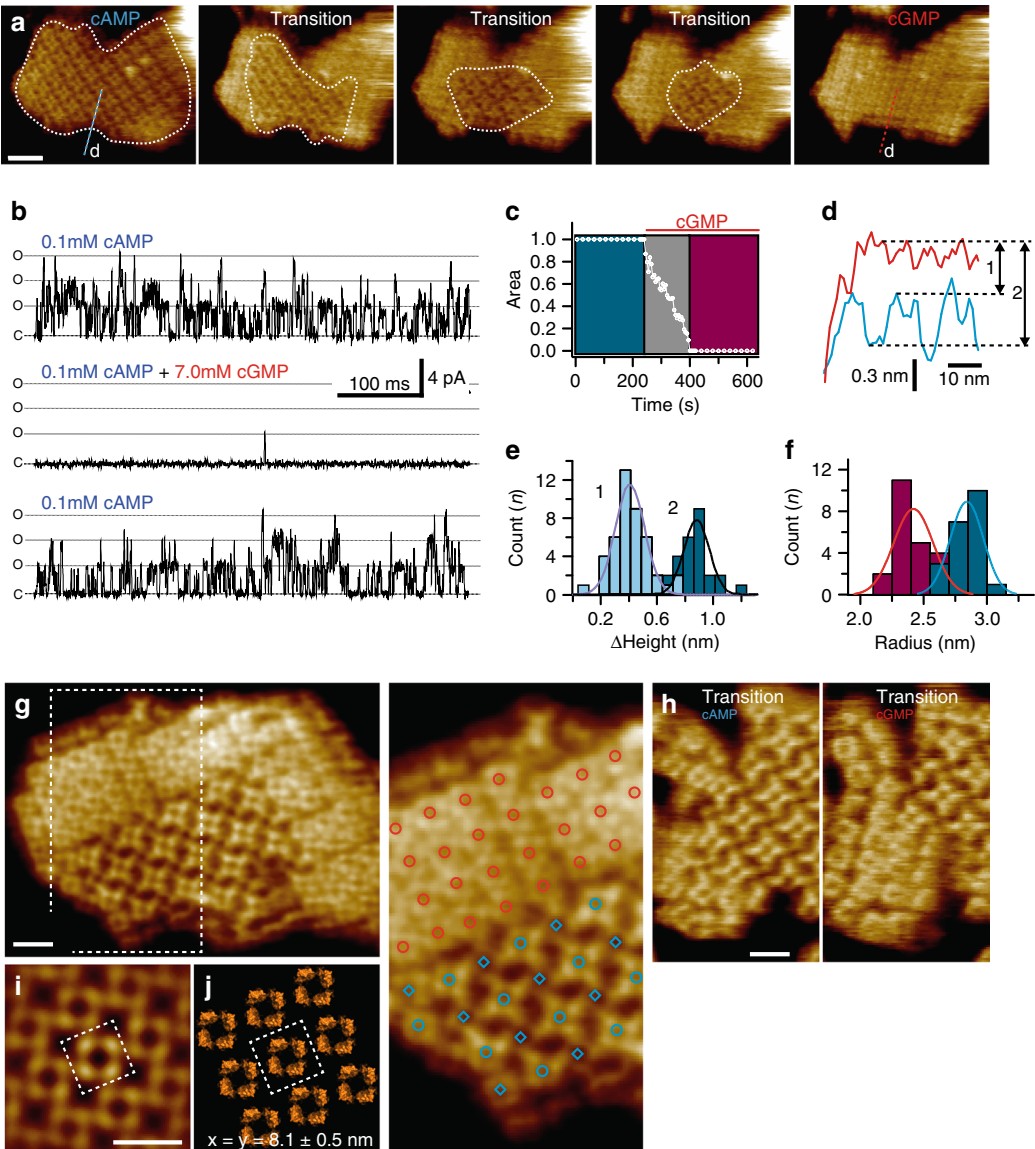

**Fig. 3** Dynamics of ligand-induced conformational changes in SthK by real-time HS-AFM imaging. **a** HS-AFM time-lapse high-resolution image sequence of a SthK 2D-crystal initially in 0.1 mM cAMP and exposing CNBDs. Upon addition of 7 mM cGMP, SthK channels undergo a conformational change progressively from the borders to the center of the membrane patch (dotted white outline). Scale bar: 30 nm. **b** Representative electrophysiology traces from SthK channels in the presence of 0.1 mM cAMP (top and bottom traces), and 7 mM cGMP and 0.1 mM cAMP (middle trace). Activity from the same bilayer, perfused to different solutions is displayed to enable direct comparisons. C and O are closed and open channel levels, respectively. Three channels appear active, each with a Po of ~0.25 at 0.1 mM cAMP. **c** Kinetics of cAMP to cGMP conformational transitions of the 2D-crystal shown in **a**. **d** Height profiles along the blue (cAMP) and red (cGMP) dashed lines in **a**. **e** Distributions of the relative height differences of the very same molecules in the cGMP conformation and the higher (1) and lower (2) molecules in the cAMP conformation. **f** Histograms of the radial distance of the cAMP (blue) and cGMP (red) CNBDs from the central four-fold axis. **g** Left: High-resolution topography of a membrane containing well-ordered channels in both conformations. Right: Zoom into region outlined in **g** with overlaid red circles for the resting state channels and blue circles (upper molecules) and blue rhomboids (lower molecules) in the activated state. The crystal was imaged in the presence of 0.1 mM cAMP and was one of a handful of membranes that contained resting and activated state molecules simultaneously (scale bar: 10 nm). **h** High-resolution topographs during a cAMP to cGMP transition where the majority of the molecules are in the cAMP (left) and in the cGMP (right) conformation, respectively (scale bar: 10 nm). **i** Correlation average of the resting state 2D-crystal in **g**. Scale bar: 10 nm. **j** Packing model of the 2D-crystal using the SthK structure (PDB 6CJQ). The unit cell (dashed square in **i** and **j**, dimensions $a = b = 8.1$ nm, $\gamma = 90°$), comprises one tetramer

due to a conformational selection during the reconstitution and crystal formation, where contacts within the lattice resulted in small crystals where the molecules are stabilized in the same active conformation. Owing to the high signal-to-noise ratio of HS-AFM and their signature crystal packing, described above, the membrane patches containing active molecules could be easily identified and specifically addressed.

**SthK conformational changes are large and reversible.** We next investigated the conformational changes within the channel molecule that led to the observed dramatic packing changes from activated/open to resting/closed (Fig. 3a). Cross-section analyses of the heights of the molecules show that all cGMP-bound channels protrude further from the membrane than the cAMP-bound channels, by ~0.4 nm with respect to the higher, and

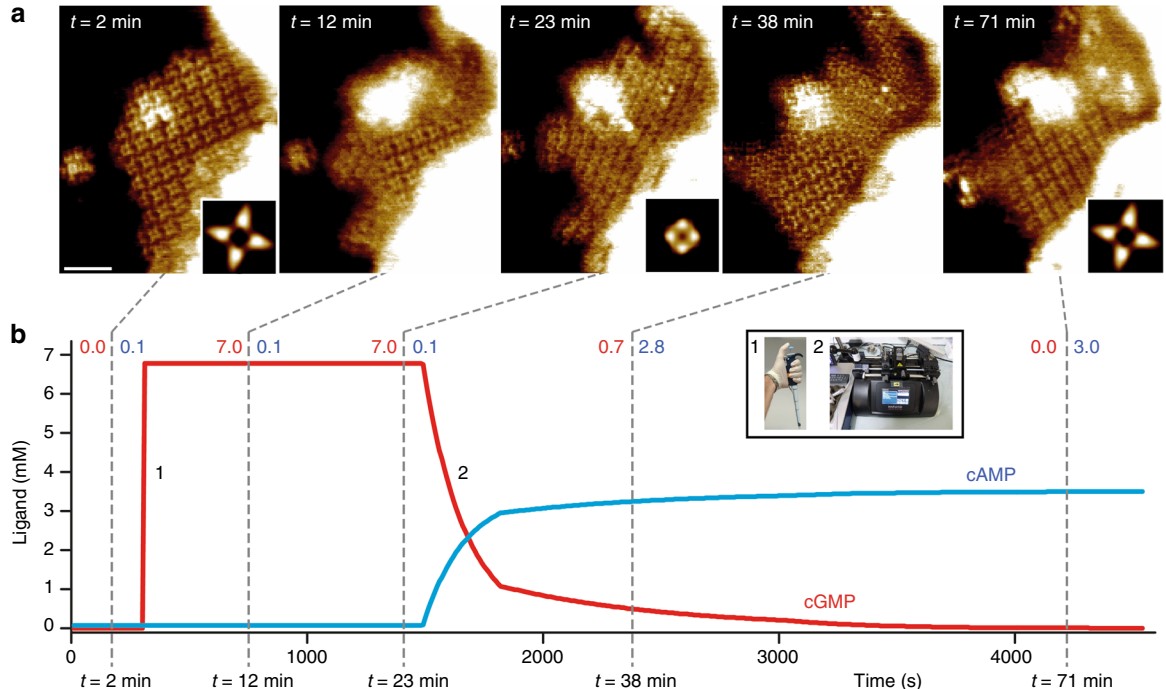

**Fig. 4** Reversibility of cyclic nucleotide-dependent conformational changes. **a** HS-AFM time-lapse high-resolution image sequence showing CNBD conformational and 2D-lattice rearrangement dynamics of SthK upon cGMP injection (panels $t = 12$ min and $t = 23$ min) and after controlled cGMP removal and cAMP re-addition via a fluid exchange pumping system (panels $t = 38$ min and $t = 71$ min). Insets in $t = 2$ min, $t = 23$ min and $t = 71$ min are correlation averages ($n = 22$, $n = 39$, and $n = 17$ tetramers) of the corresponding frames. The molecular conformational changes lead to local membrane bending (bright area in the top part of the membrane). Scale bar: 30 nm. **b** cAMP (blue) and cGMP (red) concentrations as a function of time during the HS-AFM experiment. Dashed lines and adjacent labels indicate the times (bottom) and the cAMP (blue) / cGMP (red) concentrations when the images shown on top were acquired. Inset displays cGMP addition through pipetting (1) and cAMP buffer exchange (2)

~0.8 nm to the lower level cAMP-bound molecules (Fig. 3d, e). Thus, a conservative estimate is that the CNBDs move ~0.6 nm vertically (the average of the measured 0.4 and 0.8 nm height difference) away from the membrane when they switch from the active to the resting state, although the displacement is likely closer to 0.8 nm since the lower cAMP-bound channels that are in contact to the surface on the other side reach the same height after cGMP addition (Fig. 3a; Supplementary Fig. 2c). In addition to the height changes, the four CNBDs in the tetramer move ~0.4 nm closer together, towards the channel axis, upon cGMP binding (Fig. 3f), narrowing the intracellular face entryway.

These molecular conformational changes when transiting from cAMP to cGMP likely led to the above-described striking 2D-crystal-packing remodeling, best illustrated in a 2D-crystal that contains both packing types (Fig. 3g) and high-resolution images of single molecules in conformational transition (Fig. 3h). The 2D-crystal reorganized from a lattice with unit cell $a = b = 11.6 \pm 0.7$ nm, $\gamma = 90°$ (comprising 8 SthK monomers in cAMP, see Fig. 2) to a unit cell $a = b = 8.1 \pm 0.6$ nm, $\gamma = 90°$ (comprising 4 SthK monomers in cGMP, Fig. 3g, right). This very same packing was observed when SthK channels were reconstituted in the absence of cAMP (Supplementary Fig. 3c,d). Moreover, no obvious structural differences between the cGMP-bound state and *apo* state could be detected. Correlation-averaging of the resting state 2D-crystal (Fig. 3i), was docked with the resting-state cryo-EM structure (Fig. 3j). The structural packing model thus built of the resting state nicely matches the packing of the molecules when viewed from the extracellular face (Supplementary Fig. 3d), further validating the cryo-EM SthK resting-state structure and consistent with the proposed 25° CW rotation upon activation. In the resting state packing, the VSDs of adjacent

molecules make extensive contacts in the membrane via their S3 transmembrane helices (Supplementary Fig. 4, left). Comparing the resting state packing with the activated state (cAMP) packing, the intermolecular contacts over S3 are broken or altered during state transition (Supplementary Fig. 4, right), suggesting that the ligand-induced changes may propagate to the voltage sensor domains. In support of this, in some 2D-crystals imaged during transition from cAMP to cGMP, the peripheral parts, where the channels had already undergone the conformational changes, broke off from the rest of the membrane (Supplementary Movie 2). This is suggestive of changes in the domains that make the inter-molecular contacts in the membrane, which we assumed to be the voltage-sensor domains, despite the fact that they are not visible in the AFM images (Supplementary Fig. 4).

In order to investigate whether the transition was reversible we took advantage of a fluid exchange pumping system that we have recently implemented into our HS-AFM system[27]. After first imaging the sample in the presence of cAMP (Fig. 4a, $t = 2$ min), we added excess cGMP to a final concentration of 7 mM into the fluid chamber. The same dramatic changes in the crystal packing and the CNBDs were observed, starting from the membrane border ($t = 12$ min) and moving towards the center of the patch until the change was complete ($t = 23$ min). Upon replacing cGMP with saturating cAMP (3 mM), the original morphology of the cAMP 2D-crystal was recovered ($t = 38$ min and $t = 71$ min).

## Discussion
X-ray crystallography, nuclear magnetic resonance (NMR) spectroscopy and more recently single particle cryo-EM have proven to be invaluable tools to study the structure–function relationship in ion channels. Yet, these techniques rely on ensemble averaging

and analyze the proteins in a non-physiological environment (low temperature, detergent, and absence of a membrane bilayer) providing static snapshots of fixed conformations, of which the functional state may prove difficult to assign. Here, using HS-AFM we captured the dynamics of a reversible conformational change in SthK, when the agonist cAMP is replaced by the competitive activity inhibitor cGMP and vice versa. These conformational transitions were monitored on the same molecules assembled in small 2D-crystals (Supplementary Fig. 1, Figs. 3a, 4a). Although the lattice constraints in this preparation likely affect gating energetics and transition kinetics (Fig. 3c), therefore hindering a straightforward correlation between the measured kinetics and functional data, it allowed us to capture low probability states which were overlooked in a single-particle cryo-EM study[22]. Importantly, we were able to depict the long-range conformational movements and interactions underpinning channel gating in response to cyclic-nucleotide binding.

Our real-time imaging study shows that cAMP binding induces a ~0.4 nm splaying of the CNBDs away from the central four-fold axis, compared to the more compact organization in the presence of cGMP. Furthermore, activation brings the CNBDs ~0.6 nm closer to the pore domain and the lipid bilayer. Most importantly, these two translational movements are concurrent with a ~25° clockwise rotation of the CNBDs (summarized in Fig. 5a). Electrophysiology shows that the cAMP-bound conformation leads to measurable channel activity while the cGMP-conformation is virtually closed. We therefore conclude that the conformational transitions observed with HS-AFM directly correspond to gating of the channel. The conformational changes observed here, and specifically, the overall approach of the CNBDs toward the membrane plane, are in qualitative agreement with the differences seen between the cGMP-bound, activated TAX-4 structure[17] and the apo state SthK[22], as well as previous structural studies on the prokaryotic channel MloK1[28,31,32], though MloK1 lacks the C-linker[33,34]. Moreover, a similar clockwise rotation of the C-linker and CNBD (viewed from the cytoplasm) with respect to the pore domain was previously proposed to occur in this channel family based on comparisons between the LliK structure with other channels of the family[18], as well as comparisons between cAMP-bound and apo HCN1 structures, though the structural differences were small in this case[16]. However, the extent of conformational changes observed in the static structures was limited and the inferred gating mechanism remained speculative because it was not directly monitored in real time on the same channel.

Our HS-AFM real-time imaging points towards an intuitive model for the gating mechanism. We first examined the consequences of the ~25° clockwise (viewed from the cytoplasm) rotational motion of the CNBDs on the C-linker and the pore-lining helix S6 of SthK. The four C-linkers form a tight, flat disk-shaped structure, often referred in the literature as gating ring, interfacing between and connecting the CNBDs and the pore (Fig. 5b). Importantly, the attachment of the CNBDs to the C-linker is located at the periphery of the C-linker at a radial distance $r_{(CNBD)}$ ~3 nm from the pore axis (Fig. 5b). Thus, the clockwise rotation by $\phi$~25°, induced by the CNBDs upon activation (Fig. 5b, red arrows), results in a displacement $\Delta x_{(CNBD)}$ of ~1.3 nm on the periphery of the C-linker ($\Delta x_{(CNBD)} = 2 \cdot \pi \cdot r_{(CNBD)} \cdot \phi / 360$). As a consequence, the C-linker will pull on the bundle-crossing helices S6, that form a right-handed iris diaphragm, with a clockwise torque, precisely what is needed to spread them open (Fig. 5c). Although more recently the role of S6 in CNG channel gating has been challenged[35], similar mechanisms involving a twist of the C-linkers to pull apart the S6 helices and eventually gate the channel have been suggested in earlier studies[12,36]. Given that the C-terminal

end of the bundle-crossing helix S6 is attached to the C-linker at a radial distance of ~1 nm from the axis, we propose a simple mechanical model where the C-linker acts as a mechanical lever. The conformational displacement between the peripheral attachment to the CNBD and the central attachment to the S6 helix will be divided by a factor ~3 but the torque force will be amplified by a factor ~3 (given that the sum of the momentum must be 0) (Fig. 5d). Thus each S6 would move by ~0.4 nm outward, increasing the bundle crossing diameter by ~0.8 nm and possibly opening the channel. The above estimates hold for a situation where the C-linker is mechanically very sturdy—stiffer than the S6 on which it pulls. If the C-linker were soft, the CNBDs would likely adopt a rotated and activated state without efficiently opening the pore. We know that this is not the case, based on the apo, cAMP, and cGMP cryo-EM structures[22]: The cAMP-bound structure features both a closed channel and a deactivated non-rotated CNBD. Thus we propose that coupling is strong, with a stiff C-linker connecting pore and CNBD. In essence, the lever mechanism reduces the force needed to be generated in the CNBD by a factor of ~3, and the C-linker appears to have evolved to optimize the chemo-mechanical coupling between the CNBD and the PD. Similar mechanisms have been proposed for other ligand modulated K⁺-channels such as Slo2.2 and MthK[37,38].

The structural models of the 2D-crystal packings in the resting and activated states suggest conformational changes upon ligand binding may also occur in the S1–S4 transmembrane region (Supplementary Fig. 5). In the resting-state inter-channel-contacts are mediated by residues in and around the non-helical stretch of S3, which appear to be broken upon activation (Supplementary Figs. 4 and 5). While we have no direct evidence of such rearrangements in and around S3/S4, the most intuitive interpretation for the packing changes and the observed breaking of the crystals (Supplementary Movie 2) is a ligand-induced rearrangement in the VSDs. Ligand-dependent rearrangements in the VSD have been previously established for the related, cAMP gated channel MloK1[32]. Moreover, evidence for conformational changes in the VSD in response to ligand binding has been reported for the bovine CNGA1 channels as well[39]. Provided their very little voltage sensitivity, the presence of functional voltage sensors in CNG channels has been a long-standing conundrum[40,41]. The emerging observations that VSDs may actually be involved in the ligand-gating process provides a rationale for why VSDs have been preserved during evolution from bacteria to mammals in this class of channels.

One major limitation of the present study is that the conformational transition is being investigated in a 2D-crystal. As a consequence, the dynamics of the structural changes were greatly limited by the 2D-crystal packing, and the conformational change occurred over several minutes and always started from the 2D crystal border (see Fig. 3a, c). Indeed, the inter-tetramer association energy is reduced at the lattice borders where SthK channels interact with no more than 2–3 neighbors compared to molecules deeper in the lattice, where each tetramer is constrained by the four closest surrounding neighbors (Fig. 3a and Supplementary Fig. 4). The lattice constraint exerted by the four neighbors appear to substantially affect the gating energetics and hinder independent ligand-induced conformational transition in these molecules. As the conformational change starts spreading from the borders, protein-protein contacts mediating the lattice are progressively undone, allowing the conformational change to propagate deeper into the lattice and thereby explaining the domino-like propagation of the transition. Moreover, the conformational change appears to involve a concerted transition of all four subunits simultaneously. Indeed, the fact that the conformational transition is easier for molecules at the borders that

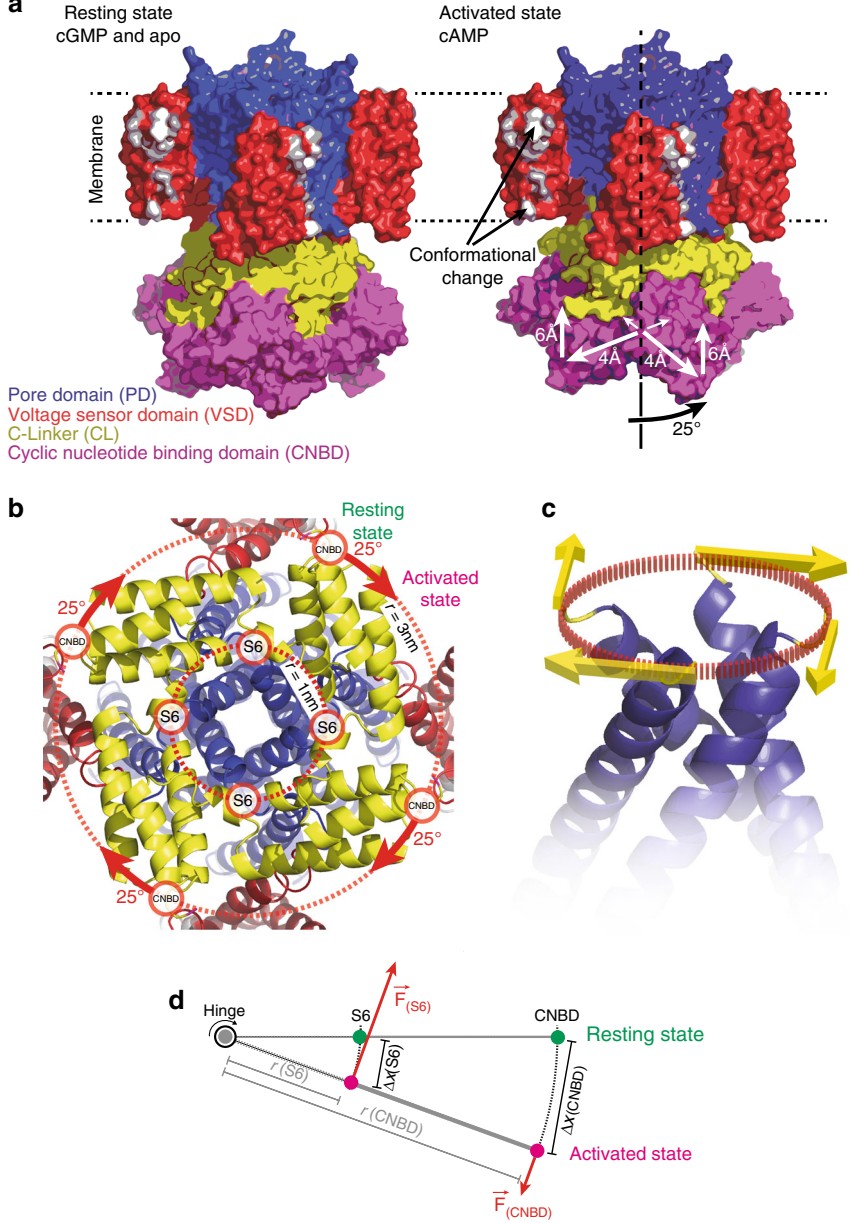

**Fig. 5** The conformational changes in SthK upon activation. **a** Left: SthK in the resting state (cartoon based on the high-resolution cryo-EM structure, PDB 6CJQ). Right: Model of SthK in the activated state: Upon activation, the CNBDs rotate by ~25° clockwise (when viewed from the intracellular side) and move by ~6 Å towards the membrane and by ~4 Å outwards from the four-fold axis (note, the activated state is a cartoon assembled using domains of the SthK structure repositioned according to the displacements found with HS-AFM). Key residues in and around S3 that are evidenced to undergo conformational changes are highlighted in white. **b** Close-up view onto the SthK C-linker (yellow) from the intracellular side in the resting state. The CNBDs are attached at the periphery to the C-linker disk with a radius ~3 nm from the central axis. Rotation of the CNBDs by ~25° clockwise induces a torque that is transmitted to the CNBD-S6 attachment. **c** Clockwise rotation of the C-linker will pull the right-handed S6 bundle-crossing open like a diaphragm. **d** Simple mechanical model where the C-linker is a type-2 lever with two attachment points to the S6 helix and to the CNBD. (S6) and (CNBD) indicate the respective attachment sites, $r$(S6) and $r$(CNBD) are the radial distance from the central axis, F(S6) and F(CNBD) the forces acting on the attachment sites, and $\Delta x$(S6) and $\Delta x$(C-linker) are the displacements, of the respective domains

have only 2–3 neighbors is an indirect indication for a concerted conformational change, where the energy barrier of all 4 protomers has to be overcome together. However, it cannot be ruled out that this is a consequence of the crystal contacts affecting the gating energetics and stabilizing the two more stable conformations (active and resting) rather than an intrinsic gating property. Intuitively one might argue that the domain-swapped architecture of the channel, and notably the fact that the C-linker forms an ensemble disk comprised of protein moiety from all four

subunits, are indicative of a fully coordinated/simultaneous structural change. All in all, the kinetic of the transition observed here cannot be compared with biochemical measurements and electrical recordings, which suggest that the channel responds to cAMP/cGMP binding in few hundreds of milliseconds[21].

In summary, we provide direct and compelling evidence that SthK channel gating is associated with a global and concerted structural transition implicating long-range interactions between distant and orthogonal protein domains, namely the CNBDs and

C-linkers to the pore domain. We provide further indirect evidence that the conformational changes also implicate the VSD. We suggest that the insight derived from this study extends to other cyclic nucleotide-modulated channels and possibly other channels modulated by binding of small intracellular ligands to carboxyl terminal regulatory domains such as $K_{ir}$, calcium activated potassium channels and several TRP family members. Moreover, our work establishes HS-AFM as a unique and exquisitely sensible tool to study directly and at single molecule level the mechanism and dynamics of partial agonists, which are difficult to capture by low signal to noise techniques relying on ensemble averaging, such as X-ray crystallography and cryo-EM.

## Methods

**Sequence alignment and phylogenetic tree construction**. Sequences were aligned using ProsmalS3D[42], constraining the alignments to the two available SthK homologues high-resolution structures in the holo-state; 5U6P resolved in the presence of cAMP (the human HCN1 channel[16]) and 5H3O (referred as TAX-4, an eukaryotic homolog of CNGA channels[17]) resolved in the presence of cGMP. A maximum likelihood phylogenetic tree was generated from this alignment using PhyML[43]. LG model, four substitution rate categories, estimated proportion of invariable sites and estimated gamma shape parameter (default options) were used. Trees were then visualized and edited at ITOL (http://itol.embl.de/)[44].

**SthK expression and purification**. The gene for the C-terminally truncated SthK 1–420 (UniProtKB G0GA88) was cloned into the pCGFP-BC vector, and further modified by deleting the GFP and four out of the 8 Histidines, as previously described[21,22]. Briefly, SthK protein was expressed in *E. coli* C41 (DE3) cells (Lucigen) grown in LB media supplemented with 100 mg/ml ampicillin, via induction with 0.5 mM IPTG for 12 h at 20 °C. The cells were harvested, sonicated and the membranes solubilized with 30 mM n-Dodecyl-β-D-Maltopyranoside (DDM, Anatrace) for 1.5 h at 4 °C. The solubilization buffer (SB) had 20 mM HEPES, pH 8.0, 100 mM KCl, 0.2 mM cAMP. During sonication PMSF (85 μg/ml), Leupeptine/Pepstatin (0.95/1.4 μg/ml), 1 mg of DNaseI (Sigma), 1 mg of Lysozyme (Sigma), and one cOmplete™ ULTRA mini Protease Inhibitor (Roche) tablet were added. The extract was spun down at 37,000 g and the supernatant was applied to a $Co^{2+}$ affinity column, which was washed with buffer SB with 1 mM DM and 40 mM imidazole. The protein was eluted with buffer SB with 1 mM DM and 250 mM imidazole. The protein was further purified on a Superdex 200 10/300 GL column (GE Lifesciences) equilibrated in running buffer (20 mM HEPES, pH 8.0, 100 mM KCl, 200 μM cAMP, 0.3 mM DDM) at room temperature. The peak fraction at ~11 ml containing SthK tetramer was collected and concentrated to 10–12 mg/ml with a 100 kDa cut-off concentrator (Amicon Ultra, Millipore). The final protein concentration was determined using the molar extinction coefficient at 280 nm: 55,900 $M^{-1}$ $cm^{-1}$.

For 2D crystallization, detergent-solubilized SthK was mixed with *E. coli* polar lipid extract (Avanti Polar Lipids) at a lipid-to-protein ratio of 0.8 to 1.0 (w:w) and dialyzed against detergent-free buffer (100 mM KCl, 20 mM Tris-HCl pH7.6, 0.2 mM cAMP) for 2D-crystallization in dialysis buttons for 5–10 days (in buffer with cAMP ligand) at 37 °C (the dialysis buffer was exchanged every other day). After crystallization and before HS-AFM analysis, the C-terminal hexahistidine-tagged was cleaved for 2–3 h at 20 °C with 200 NIH units of thrombin (Sigma) per mg of protein. To stop the reaction, 0.1 mM Pefabloc (Sigma) was added and sample was washed three times with crystallization buffer. Supernatant was removed after the 2D crystals had sedimented and fresh crystallization buffer added.

**HS-AFM imaging**. A 1 mm diameter muscovite mica plate was glued on a HS-AFM glass rod sample holder and mounted on the scanner. Reconstituted SthK were adsorbed on freshly cleaved mica for 20 min. Subsequently, if not otherwise indicated, the sample was rinsed with imaging buffer (20 mM HEPES, pH8.0, 100 mM KCl) containing 100 μM cAMP. Within the sample coexisted membrane areas densely packed with up-oriented and down-oriented SthK channels and membrane areas in which the SthK channels are crystallized in 2D-lattices (Supplementary Fig. 1a). The small 2D-crystals in the active state represented approximately ~10% of the observed sample with AFM (with large variations between different preps), with the other 90% displaying the alternating up-and-down packing, which were imaged (for technical limitations) at somewhat lower resolution and the structural state could not be unambiguously assigned. The high signal-to-noise ratio of the HS-AFM allowed us to target channels crystallized in 2D-lattices selectively and at high resolution. We discovered that only the small 2D-crystals (~10% of the sample) contained molecules that changed conformation upon cGMP addition. Although single-channel recordings indicate a low open probability at 0 mV ($Po$ = ~0.1) in the presence of saturating cAMP, we believe it is likely conformational selection of the ~10% activated state molecules that gather together in the reconstitutions. Moreover, crystal contacts in the lattice likely stabilized them in the activated state (see Discussion). The fluid exchange pumping

system was as previously described[27]. HS-AFM was operated in oscillating mode, equipped with ultra short cantilevers of 8 μm in length (USC, NanoWorld, Switzerland) with a spring constant of 0.15 Nm⁻¹, a resonance frequency of ~600 kHz and a quality factor of ~1.5 in buffer. The applied force to the sample was minimized by adjusting the free amplitude ($A_0$) to ~10 Å and the imaging amplitude setpoint ($A_s$) to ~90% (~9 Å) of the free amplitude (the force F can be estimated following $F = \{k_c[(1-\alpha) \times (A_0^2 - A_s^2)]^{1/2}\}/Q_c$, where α is the ratio of amplitude reduction by frequency shift to total amplitude reduction (typically $\alpha = 0.5$), $k_c = 0.15$ N m⁻¹ is the cantilever spring constant, and $Q_c = 1.5$ is the cantilever quality factor in liquid). Membrane areas of $100 \times 100$ nm² to $250 \times 250$ nm² were imaged at $200 \times 200$ or $300 \times 300$ pixels per frame, respectively, with scan speeds between 500 and 1000 ms per frame.

**Electrophysiology**. We reconstituted SthK into liposomes according to previous protocols[45,46]. Briefly, the purified protein in 1 mM DDM was first mixed with a combination of DOPC, POPG and Cardiolipin lipids (Avanti Polar Lipids) in the ratio of 5:3:2 solubilized with 34 mM CHAPS in a buffer containing 400 mM KCl, 10 mM HEPES, with pH adjusted to 7.6 using NMG. Subsequently, the detergent was removed by applying the mixture to a 20 ml gel-filtration column packed with Sephadex G-50 beads (GE Healthcare). The turbid liposome-containing fractions eluted from the column were aliquoted, flash-frozen in liquid nitrogen and stored in −80 °C for electrophysiological measurements.

In order to examine the response of SthK to different ligands, a horizontal planar lipid bilayer system consisting of a *trans* (bottom) and *cis* (top) chamber separated by a plastic partition was used[47]. The chambers are connected to an Axopatch 200 A amplifier (Molecular devices) via agar bridges filled with a solution containing 150 mM KCl, 5 mM EDTA, with pH adjusted to 8 using KOH. Bilayers were formed over the partition hole (~100 μm in diameter) by application of 6.25 mg/ml DPhPC lipids dissolved in *n*-decane using a fire-polished capillary glass tube. During electrophysiological experiments, the freshly-thawed and sonicated liposomes were applied to the lipid bilayer and channel incorporation was monitored with a constant voltage of +100 mV in gap-free mode. Electrophysiology buffer contains 97 mM KCl, 3 mM KOH, 10 mM HEPES, pH7.0. To achieve ligand exchange, the *trans* chamber was perfused with at least 5 times the chamber volume of the electrophysiology buffer that contains cAMP or cGMP at the concentrations specified in the text. All current signals were acquired with Clampex (V10, Molecular Devices), filtered at 1 kHz with an 8-pole low pass Bessel filter and digitized at 20 kHz with Digidata 1440 A digitizer (Molecular Devices). Data were analyzed in Clampfit (V10, Molecular Devices) where single-channel recordings were idealized with the single-channel search mode The number of channels was estimated directly from the recordings at saturating cAMP and +100 mV, by dividing the largest current amplitude in the recording (where all channels in the bilayer are open at the same time) by the single-channel current amplitude. Alternatively, one can count the number of steps, as shown in Fig. 3b by the three dashed lines labelled o, indicating the levels where one, two, and all three channels are open at the same time, respectively. Since the single-channel open probability under these conditions is large enough that the probability of having all channels in the bilayer open at the same time is large, this value gives an accurate estimate of the number of channels. The average single channel open probability at saturating cAMP concentrations and +100 mV ($Po$ ~ 0.4) has been previously determined from a large number of recordings where only single channels were present in the membrane[21]. Traces are filtered offline to 500 Hz for display. Open probabilities are presented as mean ± s.e.m.

**AFM data analysis**. HS-AFM images were first-order flattened and contrast adjusted using laboratory-made routines in Igor Pro software (WaveMetrics, Lake Oswego, OR, USA). The movies were then drift corrected, by means of frame-to-frame cross-correlation, using a lab-made image analysis software plug-in for ImageJ[48,49]. Average topography of the movies were calculated using the standard measurement tools in ImageJ. Symmetrized, correlation-averaged topographies were calculated using a lab-made image analysis software plug-in for ImageJ[49]. No statistical methods were used to predetermine sample sizes, but our sample sizes are similar to those reported in previous publications[25–28]. We excluded low lateral resolution data from analysis or data where the resolution was not stable and deteriorated during data acquisition.

**Structural models of the 2D-crystal packings**. We used the resting state SthK structure (PDB 6CJQ) to build models of the supramolecular arrangement of the SthK channels in the 2D-crystals[50]. To improve accuracy, we used correlation-averages of the HS-AFM images using average unit cell dimensions: activated state unit cell $a = b = 11.6 \pm 0.7$ nm, $\gamma = 90°$, comprising 8 SthK monomers and resting state unit cell $a = b = 8.1 \pm 0.6$ nm, $\gamma = 90°$, comprising 8 SthK monomers. The unit cell dimensions determine the lateral position of the channels. To determine the rotational degree of freedom, the HS-AFM topographies resolved well the turrets between the S5 and pore helix on the extracellular face and the CNBDs on the intracellular face allowing to determine the center-of-mass, i.e., the integrated volume under these protrusion, of these features and them being best fitted with the cryo-EM structure. Finally, the vertical shift of every

second molecule in the activated state was set to 0.3 nm as determined by the cross-section analysis.

## Data availability

Data supporting the findings of this manuscript are available from the corresponding authors upon reasonable request.

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

## Acknowledgements

This work was supported in part by National Institutes of Health grants (R01GM124451) to C.N. and S.S. Early parts of the work were funded by a European Research Council (ERC) Grant (#310080, MEM-STRUCT-AFM) to S.S. A.M. was supported by a Long Term EMBO Fellowship (ALTF 1427-2014) and a Marie Curie Action (MSCA IF-2014-EF-655157).

## Author contributions

A.M., S.S. and C.N. designed the research; A.M. performed AFM measurements and X.G. performed electrophysiology; S.S. and A.M. performed modeling and structural analysis. A.M. and S.S. analyzed data from AFM experiments; X.G. and C.N. analyzed data from

electrophysiological experiments; R.A. and J.R. purified and reconstituted protein. A.M., S.S. and C.N. wrote the manuscript with input from all authors.

## Additional information

**Competing interests:** The authors declare no competing interests.

