## [Peer Review File · Nature Communications]

Reviewers' comments:

Reviewer #1 (Remarks to the Author):

This paper uses high-speed atomic force microscopy (HS-AFM) to visualize the activation conformational change in a bacterial cyclic nucleotide-gated (CNG) channel called SthK. The authors reconstitute purified SthK channels in lipid bilayers and probe their surface structure from the extracellular and intracellular sides, in the presence of the agonist cAMP, the competitive antagonist cGMP, and in the apo state. They find the channels from well-ordered 2D crystals in the membrane. In the presence of cAMP, the configuration is incompatible with their recent, unpublished, cryo-EM structure of SthK, and suggestive of a 25° rotation and splaying of the cyclic nucleotide-binding domains (CNBDs) upon cyclic nucleotide activation. In the presence of the competitive antagonist cGMP (and in the apo state) the channels rearrange to produce a structure more like their cryo-EM structure. In addition the crystal packing changed from which the authors infer there is a change in the voltage-sensor domain (VSD). They conclude that there is an iris diaphragm-like motion of the CNBD and C-linker regions that gate the channel.

The paper clearly establishes HS-AFM as a powerful method for measuring structural rearrangements in channels. Combined with known high resolution structures, HS-AFM can reveal large structural rearrangements in a near native environment and measure the time course of the rearrangements (albeit with limited time resolution). The structural rearrangement proposed is essentially the same as that proposed from multiple cryo-EM structures of related CNG and HCN channels (the similarity to the open TAX-4 cryo-EM structure should be discussed more thoroughly in the manuscript). One potential issue is that there does not seem to be evidence that deactivation in the presence of cGMP involves a rotation of the CNBDs back to the position relative to the PD that they observe in the cryo-EM structure. Did you also image the cGMP-bound crystal form from the extracellular side to directly compare the position of the CNBDs relative to the PD? This seems like an important measurement to establish that the rotation observed is not simply a rotation of the whole channel. Similarly, the proposed 6 Å vertical movement seems arbitrary (chosen because it is half way between the lower and upper positions of the transmembrane domain with cAMP bound). Once again, an image of the cGMP-bound crystal from the extracellular side would establish the size of the vertical movement. In addition there are a number of other points that need to be addressed:

1) The advantages of HS-AFM are overstated and need to be toned down. SthK is being studied in a crystal, which is itself non-physiologic and involves crystal packing interactions which probably perturb the protein energetics (see below). The time resolution also isn't high enough where you can actually measure the kinetics of the transition. And, the functional state of the conformation in each of the lattices is impossible to assign because there is no way of knowing the conformation of the pore. The big advantage of HS-AFM seems to be that the structures in the resting and activated states can be determined on the same channels.

2) Page 5 "...comparison between the 2D-crystal packing model of the resting state...revealed that the CNBDs assume a different conformation than that seen in the cryo-EM structure." A comparison should be made between the proposed cAMP-bound structure and previously determined cryo-EM structures (i.e. Tax4, LliK, HCN1)?

3) Page 7: "...the open probability decreased ~1000 fold from 0.25 to 0.0003". How was the number of channels estimated in Figure 3b to determine the open probability? Is the open probability the same at 0 mV as at 100 mV? In addition, why are all of the channels activated in the HS-AFM measurements with cAMP but only 25% of them are open with electrophysiology? Could crystal contacts be effecting the energetics? More discussion is required.

4) Page 7: "...broke off from the rest of the membrane suggesting that the conformational rearrangements...are important" How does breaking of the crystal suggest importance? The VSDs are not visible with AFM, and simply changes in crystal contacts do not suggest movement. The speculation about movement of the VSD should be removed.

5) This discussion of forces contains a number of inaccurate assumptions and provides no new insights. For example, there is no reason to think that the pulling force for opening CNG channels will be similar to that of voltage-activated channels. The $\Delta\Delta G$ for voltage-gated channels is much larger than for CNG channels. This discussion should be removed.

Minor comments:

6) Page 13. "...lipid-to-protein ratio of 0.8–1.0..." Is this molar ratio or wt:wt ratio?

7) Figure S2. Is the bottom green layer the mica surface? It is labeled "lipid bilayer". Also, in the caption, "CNNB domain" should be "CNBD".

8) Page 3. "...the conformational changes upon ligand binding are minor in this channel 16" This should be reference 17, not 16.

Reviewer #2 (Remarks to the Author):

In this study, the authors observed dynamic structural changes in one of cAMP-gated ion channels, SthK, during the interconversion between the resting (apo and cGMP) and the activated (cAMP) states, using high-speed atomic force microscopy (HS-AFM). They succeeded in capturing at 1 frame/sec different structures between the two states and propagation of the structural changes over SthK molecules in the 2D crystals. From the observed results of structural changes, the authors proposed a model for gating mechanism in this channel protein.

This study obtained interesting results, in particular the propagation of conformational changes over a wider area. However, the following issues have to be clarified before publication in Nature Communications.

1. In the cryo-EM studies of SthK performed by a co-author group (manuscripts were submitted; Refs 21 & 22), they solved three structures of SthK in apo, cGMP-bound and cAMP-bound states. However, all these structures were in the resting state. This is very peculiar because the HS-AFM study clearly captured the structure in the activated state (open-state structure) in the presence of cAMP. Was the open-state structure observed at a low probability also in the HS-AFM experiments? How low was the probability? Did the probability prohibit them from analysing the open-state structure by cryo-EM? Are there other reasons for the failure of determining the open-state structure by cryo-EM?

2. The authors mentioned that SthK showed the open-state structure only in the ordered 2D-crystals but not in the alternating up-and-down packing. What is the appearance frequency ratio between the two differently packed systems? Was the alternating up-and-down packing also observed by cryo-EM?

3. The advantage of HS-AFM was not taken in the following argument: "during activation, the CNBD undergoes a 25° CW rotation compared to the resting state". Rather than comparing AFM images in the open state to the cryo-EM structure in the resting state, this rotation should be (and can be) directly shown by HS-AFM imaging.

4. According to Ref. 20, SthK selects K⁺ over Na⁺. Therefore, it must be interesting to observe SthK also in the presence of Na⁺, to know whether Na⁺ cannot flow the channel even in the open

structure or whether the channel cannot open in the presence of Na⁺.

5. Why did not the authors use photolysis of caged cAMP and cGMP? It takes time to replace the solution by a syringe pump. In the method the authors used, it is difficult to know when the concentration of cGMP reaches its maximum and how fast the structure changes upon binding to cAMP or cGMP. Flash photolysis of caged cAMP and cGMP has been successfully used in previous electrophysiological studies of SthK.

6. The authors observed propagation of changes in the 2D crystal packing and protein structure towards a wider area. Since the area of 2D crystal is very small, the concentration of cGMP or cAMP must be uniform over the area. This propagation seems to be a most interesting finding in this study. The authors should discuss this domino-like phenomenon more extensively.

7. The open and closed state structures were captured by HS-AFM imaging. However, the structure of intermediate states were not shown at the single-molecule level. As the authors mentioned, the HS-AFM system allows the observation of single molecules at ~100 ms temporal resolution. Nevertheless, all HS-AFM images were captured at 1 frame/sec. Have the authors attempt to image SthK at 10 frames/sec? If not, it should be done.

8. Does the structural change in individual SthK molecules occur simultaneously for all four subunits within a molecule? This may be able to be answered by imaging at a higher rate.

8. No AFM images were shown for SthK in the apo state. They should be shown. Is there any structural difference between the apo state and the cGMP-bound resting state?

9. Are the rate constants for on/off of cAMP and cGMP known? If known, the authors should discuss the observed structural changes in relation to these rate constants.

10. In Figure 3g, two different packing arrangements are seen within one 2D crystal patch. Considering the small area, the concentration of cGMP must be uniform over the entire area and all the molecules must have bound cGMP. Nevertheless, two distinct regions appear in one 2D crystal. What is the mechanism underlying this block separation between the two states?

Minor points

11. In Figure 4a, regions having very white (large height) objects appear. What are these objects?

12. Figure 1c shows the top view from the cytoplasmic side (Top) and the bottom view from the extracellular side (Bottom). However, in the side view the top is the extracellular side, while the bottom is the cytoplasmic side. These must be coincided with each other to avoid confusion.

Reviewer #3 (Remarks to the Author):

Report on manuscript „An iris diaphragm...“ by Marchesi et al.

The present paper reports on an interesting set of experiments in which high speed AFM imaging is used to monitor the dynamics of global conformational changes in a cyclic-nucleotide gated channel. The authors show that the presence and absence of a ligand causes a concomitant rotational and upward movement of the cytosolic domain with respect to the part of the channel, which is harbored in the membrane. Even though this reviewer is not an expert of AFM measurements the data and the quantitative analysis appear convincing. In the field of ligand gated channels these are important data, because they finally provide some insights into long-range interactions and long-range conformational movements. This is important for an

understanding of a communication between the ligand binding domain and the channel pore. For this reason I support publication of the manuscript.

But before going into print the authors should first revise their manuscript with respect to a better presentation/discussion.

I strongly suggest the authors to restrict their interpretation to the gating of CNG channels. One of my main concerns of the paper is an over-interpretation of the data with respect to an invalid extrapolation to HCN channel gating. Other than stated in the 1st paragraph of the introduction, CN binding opens CNG channels but not HCN channels. HCN channels are opened by voltage and CN binding is only lowering the energetic barrier for activation. The gating mechanism of HCN channels cannot be discussed without information on the movement of the voltage sensor. The present data give a hint on the dynamics of the CNBD in HCN channels in response to ligand binding/dissociation. They do not explain gating of HCN channels.

The authors should discuss their data in the context of previous studies, which already proposed a rotational movement of the CNBD \pm cAMP. Craven et al.. (2008) Johnson & Zagotta 2001. The latter reference is cited in the text but in a different context.

I do not understand the conditions of Fig. 2. The authors seem to observe the channel in the same preparation in two conformations. I don't see from the text if the protein was examined in the absence or presence of cAMP. Does the co-existence of both forms imply that the cAMP concentration (if present) is not saturating.

The information in Fig. 2 i and j is very important but difficult to see because of the colors in the images. May be the authors can improve the visibility of the rotational displacement between the proteins.

Reviewer 1

This paper uses high-speed atomic force microscopy (HS-AFM) to visualize the activation conformational change in a bacterial cyclic nucleotide-gated (CNG) channel called SthK. The authors reconstitute purified SthK channels in lipid bilayers and probe their surface structure from the extracellular and intracellular sides, in the presence of the agonist cAMP, the competitive antagonist cGMP, and in the apo state. They find the channels from well-ordered 2D crystals in the membrane. In the presence of cAMP, the configuration is incompatible with their recent, unpublished, cryo-EM structure of SthK, and suggestive of a 25° rotation and splaying of the cyclic nucleotide-binding domains (CNBDs) upon cyclic nucleotide activation. In the presence of the competitive antagonist cGMP (and in the apo state) the channels rearrange to produce a structure more like their cryo-EM structure. In addition the crystal packing changed from which the authors infer there is a change in the voltage-sensor domain (VSD). They conclude that there is an iris diaphragm-like motion of the CNBD and C-linker regions that gate the channel.

The paper clearly establishes HS-AFM as a powerful method for measuring structural rearrangements in channels. Combined with known high resolution structures, HS-AFM can reveal large structural rearrangements in a near native environment and measure the time course of the rearrangements (albeit with limited time resolution).

Response: We thank the reviewer for this overall positive assessment of our work.

The structural rearrangement proposed is essentially the same as that proposed from multiple cryo-EM structures of related CNG and HCN channels (the similarity to the open TAX-4 cryo-EM structure should be discussed more thoroughly in the manuscript).

Response: We extended our discussion concerning the similarity of our findings with comparisons between SthK and TAX-4 cryo-EM structures (see second paragraph of Discussion, page 12).

One potential issue is that there does not seem to be evidence that deactivation in the presence of cGMP involves a rotation of the CNBDs back to the position relative to the PD that they observe in the cryo-EM structure. Did you also image the cGMP-bound crystal form from the extracellular side to directly compare the position of the CNBDs relative to the PD? This seems like an important measurement to establish that the rotation observed is not simply a rotation of the whole channel.

Response: As suggested by the reviewer, in order to establish the orientation of the CNBDs relative to the PD in agreement with the closed state cryo-EM structure, we have imaged the SthK crystals exposing the extracellular face (PD) in the resting state (see below). Importantly, the structural model of the 2D packing built by docking the resting state cryo-EM structure into the resting state CNBDs in the presence of cGMP (see Fig. 3i,j) match the packing of the molecules in these new images when viewed from the extracellular face (new panels in Fig. S3). Conversely, the resting state cryo-EM structure cannot be docked into the active, cAMP-bound state lattices, and required a ~25° rotation of the CNBDs with respect to the pore domain (see Fig. 2h,i). We also detail the above thoughts further in the results section of the revised version (Page 9, top).

Similarly, the proposed 6 Å vertical movement seems arbitrary (chosen because it is half way between the lower and upper positions of the transmembrane domain with cAMP bound). Once again, an image of the cGMP-bound crystal from the extracellular side would establish the size of the vertical movement.

Response: The proposed 6Å vertical movement was not chosen arbitrarily and we have now improved the description of the analysis in the text of the results section (Page 8, bottom half). Taking the example of figure 3a during transition. We are analysing delta-heights of the same membrane-embedded molecules upon ligand exchange: The resting state (cGMP) molecules are all at the same height (level) in the membrane compared to the activated state (cAMP) molecules, which are packed in a height-staggered way (high-low-high-low). So, in order to compare the resting and activated molecules, we subtract the height of the activated molecules, which have two different height levels, from the height of the resting molecules in the same membrane, and we get two different values for the difference (8Å and 4Å). We preferred to provide the middle, 6Å, as a conservative estimate of the displacement. Likely, the displacement is closer to 8Å because the activated particles with the lower height on the intracellular face are further out on the extracellular face (see figure 2a). Therefore, the lower-height molecules likely make the contact with the support, similar to the resting state molecules. We state that 6Å is a conservative estimate but the displacement is more likely 8Å, in the revised version.

To address the suggestion of the reviewer that imaging of the extracellular face would help establishing absolute height of the molecule in order to compare the two states, we would like to stress that the ~1-2 Å resolution in z-dimension typically achieved by AFM is only this precise when relative heights are measured in the same sample (as we do here). Absolute height measurements - such as those obtained from different preparations on different days etc - are expected to be affected by systematic errors owed to different z-piezo drift and creep and thermal extensions.

In addition there are a number of other points that need to be addressed:

1) The advantages of HS-AFM are overstated and need to be toned down. SthK is being studied in a crystal, which is itself non-physiologic and involves crystal packing interactions which probably perturb the protein energetics (see below). The time resolution also isn't high enough where you can actually measure the kinetics of the transition. And, the functional state of the conformation in each of the lattices is impossible to assign because there is no way of knowing the conformation of the pore. The big advantage of HS-AFM seems to be that the structures in the resting and activated states can be determined on the same channels.

Response: We agree. Consequently, we have revised the manuscript to tone down the importance of the use of HS-AFM in this study (end of Intro: page 3 bottom, Results: page 8 top, Discussion: page 12 top and page 14). We now explicitly state in the discussion that the 2D-crystals indeed affect protein energetics, substantially “slowing down” the kinetic of the active-to-resting state interconversions and hindering a straightforward correlation between the measured kinetics and the “real” protein kinetics, which was anyway beyond the scope of this manuscript and not attempted here. As pointed out by the reviewer, another limitation is that the structural information is limited to surfaces, and this is now clearly stated as well in the introduction. However, besides the possibility to image structural changes in the very same molecules, significant advantages do still exist. We think that the reviewer would agree with us that 2D-crystals are still more physiological than 3D-crystals (X-ray diffraction) and preferable to the detergents solubilized (single-particle cryo-EM) preparations. Finally, the analysis is done at ambient temperature and pressure preferable to 100K (X-ray crystallography and cryo-EM).

Also, we would like to mention that despite the fact that each image is taken at 1s imaging speed, these images contain about 300 scan lines (3.3ms) and thus each molecule described by about 15 scan lines actually existed as recorded for ~50ms. Thus, while the transition in the 2D crystal is indeed significantly slowed down (~150s, figure 3c), the transition can at least be observed (Results: page 8 top).

2) Page 5 “...comparison between the 2D-crystal packing model of the resting state...revealed that the CNBDs assume a different conformation than that seen in the cryo-EM structure.” A comparison should be made between the proposed cAMP-bound structure and previously determined cryo-EM structures (i.e. Tax4, LliK, HCN1)?

Response: This is a good suggestion, and we have expanded our discussion to discuss our data in context to CNG channel gating and the Tax-4 structure (Discussion: page 12). It is notable that some of the authors just published several SthK cryo-EM structures, and this paper features a structural comparison with the structure of the activated state of Tax-4 (Rheinberger et al., eLife 2018). However, it is important to highlight that the key element to determine the CNBDs 25° CW rotation was the docking of the SthK structural model to the extracellular AFM topography. In these images, the only visible structures are the loops connecting the S5 helix to the P-helix, which are protruding few Å out of the membrane plane. Unfortunately, these loops are poorly conserved among SthK, HCN1 and tax4 structures (see alignment below) and significantly longer in these channels compared to SthK. Therefore, the PD of these channels (HCN/TAX4/LliK) cannot be docked with sufficient confidence to the AFM data in order to establish differences/similarities in the orientation of the CNBDs in respect to the PD.

On the other hand, the overall approach of the CNBDs towards the membrane plane is in excellent agreement with what has been observed upon comparing the TAX-4 activated state structure with the SthK resting state structure (Rheinberger et al., eLife 2018), as well as the tendency of HCN1 for a clockwise rotation upon cAMP binding (we mention this in the text, Discussion: page 12).

Partial sequence alignment among different cyclic nucleotide-modulated channels: The region between the S5 helix and the pore helix is unfortunately quite variable.

3) Page 7: "...the open probability decreased ~1000 fold from 0.25 to 0.0003". How was the number of channels estimated in Figure 3b to determine the open probability? Is the open probability the same at 0 mV as at 100 mV? In addition, why are all of the channels activated in the HS-AFM measurements with cAMP but only 25% of them are open with electrophysiology? Could crystal contacts be effecting the energetics? More discussion is required.

Response: *The open probability has been determined by analysing the single channel recordings using the single-channel search module in Clampfit 10.0 software (Molecular devices), as mentioned in Methods. The number of channels is estimated directly from the recordings at saturating cAMP and +100 mV, by dividing the largest current amplitude in the recording (where all channels in the bilayer are open at the same time) by the single-channel current amplitude (or by counting the number of steps indicated in Fig. 3B by the three dotted lines labelled "o", indicating the three open levels). Since the single-channel open probability under these conditions is large enough that the probability of having all channels in the bilayer open at the same time is large, this value gives an accurate estimate of the number of channels. The open probability at 0mV ($P_o \sim 0.1$) is different than at +100mV ($P_o \sim 0.25$), since SthK is voltage-dependent and its open probability increases with depolarization (this analysis is presented in Schmidpeter et al., JGP 2018). We amended the Fig. 3 caption and the Methods to clarify this. We speculate that conformational selection during the reconstitution in the presence of cAMP and crystal contacts within the lattice resulted in small crystals, in which all molecules are in the same active conformation. These small crystals however represented approximately 10% of the observed sample (with large variations between different preps), with the other 90% displaying an alternating up-and-down packing as we describe in Fig. S1a. This is now mentioned in the Methods section (page 17) as well as Fig. S1 caption.*

4) Page 7: "...broke off from the rest of the membrane suggesting that the conformational rearrangements...are important" How does breaking of the crystal suggest importance? The VSDs are not visible with AFM, and simply changes in crystal contacts do not suggest movement. The speculation about movement of the VSD should be removed.

Response: *As recommended by the reviewer, we have substantially downgraded our speculations in the discussion about VSD movement, and made it clear that our data do not prove such changes (page 14). However, it is important to point out that ligand-dependent rearrangements in the VSD have been previously established for the related, cAMP gated channel MloK1 (Kowal et al., Nat Comm 2014). Evidence for conformational changes in the VSD in response to ligand binding are also starting to emerge for the bovine CNGA1 channels (Maity et al., Nat Comm 2015). While we agree with the reviewer that our data do not provide direct evidence for such conformational changes in SthK channel, our data are consistent with these earlier observations for this class of channels. The packing models we constructed based on the imaged extracellular face of the channels are unambiguous and show that protein-protein contacts in the 2D crystals are mediated by the peripheral VSDs (Fig. S3 and S4). The simplest interpretation for the packing changes and the observed crystal breaks is a ligand-induced rearrangement of the VSDs. We believe that is good scientific practice that in the Discussion the authors present their interpretation of the data. Therefore, although the evidence is indirect, we think that some discussion about this aspect of our experimental data in the context of previous literature is still warranted.*

5) This discussion of forces contains a number of inaccurate assumptions and provides no new insights. For example, there is no reason to think that the pulling force for opening CNG channels will be similar to that of voltage-activated channels. The $\Delta\Delta G$ for voltage-gated channels is much larger than for CNG channels. This discussion should be removed.

Response: We have removed the mentioned section.

Minor comments:

6) Page 13. "...lipid-to-protein ratio of 0.8–1.0..." Is this molar ratio or wt:wt ratio?

7) Figure S2. Is the bottom green layer the mica surface? It is labeled "lipid bilayer". Also, in the caption, "CNNB domain" should be "CNBD".

8) Page 3. "...the conformational changes upon ligand binding are minor in this channel 16" This should be reference 17, not 16.

Response: We have corrected these errors. Lipid-to-protein ratio is w:w. The bottom green layer in Fig. S2 is indeed a lipid bilayer, positioned in between the mica and the bilayer where the channels are inserted. We indeed meant reference concerning the cryo-EM structures of HCN1 in apo and cAMP-bound conformations (Lee et al., Cell 2017)

Reviewer 2

In this study, the authors observed dynamic structural changes in one of cAMP-gated ion channels, SthK, during the interconversion between the resting (apo and cGMP) and the activated (cAMP) states, using high-speed atomic force microscopy (HS-AFM). They succeeded in capturing at 1 frame/sec different structures between the two states and propagation of the structural changes over SthK molecules in the 2D crystals. From the observed results of structural changes, the authors proposed a model for gating mechanism in this channel protein.

This study obtained interesting results, in particular the propagation of conformational changes over a wider area. However, the following issues have to be clarified before publication in Nature Communications.

1. In the cryo-EM studies of SthK performed by a co-author group (manuscripts were submitted; Refs 21 & 22), they solved three structures of SthK in apo, cGMP-bound and cAMP-bound states. However, all these structures were in the resting state. This is very peculiar because the HS-AFM study clearly captured the structure in the activated state (open-state structure) in the presence of cAMP. Was the open-state structure observed at a low probability also in the HS-AFM experiments? How low was the probability? Did the probability prohibit them from analysing the open-state structure by cryo-EM? Are there other reasons for the failure of determining the open-state structure by cryo-EM?

Response: From electrophysiological measurements, the open probability in the presence of saturating cAMP at 0mV was determined to be ~0.1, and the open state is very short-lived (sub-milliseconds); the P_o is higher and the open state is longer when depolarizing voltage is also applied (see Schmidpeter et al., JGP 2018). In both experimental systems, cryo-EM and HS-AFM, no transmembrane voltage can be applied. Therefore, even in the presence of agonist at saturating concentration and in the absence of voltage, only a small fraction of the channels (<10%) are expected to be in the active state without application of voltage.

In the single-particle cryo-EM study, due to particle picking and averaging, it is difficult to assess low-probability, short-lived conformations because one typically merges the majority of 'best' molecules; unfortunately, despite our best efforts, we were not able to identify the active state. In the HS-AFM study here, the small 2D-crystals in the active state represented approximately ~10% of the observed sample with AFM (with large variations between different preps), with the other 90% displaying an alternating up-and-down packing (Fig. S1a), which were imaged (for technical limitations) at somewhat lower resolution and the structural state could not be unambiguously assigned. We discovered that only the small 2D-crystals (~10 % of the sample) contained molecules that changed conformation upon cGMP addition. We believe it is likely conformational selection of the ~10% activated state molecules that gather together in the reconstitutions, and the high signal-to-noise ratio of the AFM that allowed us to 'select' them for investigation of the conformational dynamics. We have added text to the methods section (page 16) and results (page 8) clarifying this point.

2. The authors mentioned that SthK showed the open-state structure only in the ordered 2D-crystals but not in the alternating up-and-down packing. What is the appearance frequency ratio between the two differently packed systems? Was the alternating up-and-down packing also observed by cryo-EM?

Response: As mentioned above, roughly 10% of the sample displayed the open-state structure and reconstituted in small 2D crystals. However, the reviewer probably knows that AFM is a poor technique for screening, thus we cannot assign a percentage with certainty to the occurrence of this state in HS-AFM. We speculate that the crystal contacts in the lattice stabilized the activated state. Furthermore, the high signal-to-noise ratio of the HS-AFM allowed us to target those membranes specifically (page 8). We have never performed cryo-EM on 2D crystals.

3. The advantage of HS-AFM was not taken in the following argument: “during activation, the CNBD undergoes a 25° CW rotation compared to the resting state”. Rather than comparing AFM images in the open state to the cryo-EM structure in the resting state, this rotation should be (and can be) directly shown by HS-AFM imaging.

Response: Indeed, the high-resolution panels in 3h indicate directly a clockwise rotation of the CNBDs upon activation when comparing single molecules during transition. However, difficulties to determine the angular rotation change precisely from a simple comparison of the very same molecules in such images during transition (cAMP versus cGMP, as in Fig. 3h) stem from the impossibility to unequivocally separate the rotation of the CNBDs from the rearrangements (rotation) of the molecules as a whole. In response, we added additional figure panels in the supplementary material where we present the resting state packing in the revised version (Fig. S3).

4. According to Ref. 20, SthK selects K⁺ over Na⁺. Therefore, it must be interesting to observe SthK also in the presence of Na⁺, to know whether Na⁺ cannot flow the channel even in the open structure or whether the channel cannot open in the presence of Na⁺.

Response: The SthK channel does not conduct Na⁺ ions. It is selective for K⁺ against Na⁺, as also indicated by the signature sequence for K⁺ present in the selectivity filter of this channel (GYG). Na⁺ is only expected to affect the conduction properties, and not the gating properties of this channel (the open-closed equilibrium is likely not affected by Na⁺). Since no conformational change is expected in the channel upon changing from K⁺ to Na⁺, we do not think that imaging the channel while in Na⁺ conditions with HS-AFM would be instructive in this case.

5. Why did not the authors use photolysis of caged cAMP and cGMP? It takes time to replace the solution by a syringe pump. In the method the authors used, it is difficult to know when the concentration of cGMP reaches its maximum and how fast the structure changes upon binding to cAMP or cGMP. Flash photolysis of caged cAMP and cGMP has been successfully used in previous electrophysiological studies of SthK.

Response: The reviewer is right, and cAMP/cGMP flash photolysis has previously successfully been used to study the kinetics of other members of this channel subfamily in electrophysiology studies (we have also previously used photo-uncaging in the HS-AFM as well). However, in these experiments, the dynamics of the structural change were greatly limited by the 2D-crystal packing, and the conformational change occurred over 150s and always started from the 2D crystal border, where the molecules are less constrained (see Fig 3a). The change was slow even when bulk cGMP was directly injected with a pipette into the solution at a final ~100x higher than the reported K_d. Therefore, the kinetic of the transition observed here cannot be compared with biochemical measurements and electrical recordings, which suggest that the channel responds to cAMP/cGMP binding in few hundreds of milliseconds. Flash photolysis of caged cAMP/cGMP will be certainly of value to study channel dynamics if the preparation can be optimized to highly crowded and loosely packed patches of unidirectionally inserted channels, not assembling in 2D-crystals. Furthermore, it would implicate significantly faster image acquisition, because we know that the open-state is very short lived, even under conditions when the open probability is reasonably high (when applying voltage). We are currently working in this direction, but it is still a considerable way to go to directly couple HS-AFM with functional dynamics. We detail these thoughts in the discussion of the revised version (pages 12 top and 14).

6. The authors observed propagation of changes in the 2D crystal packing and protein structure towards a wider area. Since the area of 2D crystal is very small, the concentration of cGMP or cAMP must be uniform over the area. This propagation seems to be a most interesting finding in this study. The authors should discuss this domino-like phenomenon more extensively.

Response: The reviewer is right, and we do agree that the cAMP/cGMP concentration are uniform over the area. Experiments clearly indicate that transitions start from the periphery of the 2D-crystals, where the tetramers have fewer neighbours. Inter-tetramer association energy is reduced at the lattice borders where SthK channels interact with no more than 2-3 neighbours compared to molecules deeper in the lattice, where each tetramer is constrained by the four closest surrounding neighbours (see Fig. 3a and S4). The lattice constraint exerted by the four neighbours appear to substantially affect the gating energetics and hinder ligand-induced independent conformational transition in these molecules. As the conformational change starts spreading from the borders, protein-protein contacts mediating the lattice are progressively undone, allowing the conformational transition to propagate deeper into the lattice and thereby explaining the domino-like phenomenon. The fact that the conformational transition is easier in molecules that have only 2-3 neighbors is an indirect indication for a concerted conformational change, where the energy barrier of all 4 protomers has to be overcome together. We discussed this more extensively now in the discussion of the revised manuscript (page 14).

7. The open and closed state structures were captured by HS-AFM imaging. However, the structure of intermediate states were not shown at the single-molecule level. As the authors mentioned, the HS-AFM system allows the observation of single molecules at ~100 ms temporal resolution. Nevertheless, all HS-AFM images were captured at 1 frame/sec. Have the authors attempt to image SthK at 10 frames/sec? If not, it should be done.

Response: In this study, imaging at higher rates was not necessary, owed to the very slow kinetics of the conformational change (which take place in minutes, see Fig. 3c) in response to cGMP and cAMP addition/removal (Fig. 4). Also, we would like to mention that despite the fact that each image is taken at 1s imaging speed, these images contain about 300 scan lines (3.3ms) and thus each molecule described by about 15 scan lines actually existed as recorded for ~50ms. Thus, while the transition in the 2D crystal is indeed slow (~150s, figure 3c), the single molecules observed in these frames have been recorded during relatively short dwells (Page 8, top). This being said, we could not unambiguously detect intermediate states between the activated and resting state, suggesting that the energy landscape between the two major states is not shallow, and the interconversion from one state to the other (once the change occurs) is very rapid. This, again suggests a concerted conformational change from resting to activated, where no intermediate states are expected.

8. Does the structural change in individual SthK molecules occur simultaneously for all four subunits within a molecule? This may be able to be answered by imaging at a higher rate.

Response: This is an interesting and relevant point which warrants further investigation with HS-AFM. As mentioned above, the conformational change appears to involve a concerted transition of all four subunits simultaneously (as mentioned above, while the imaging rate is $1s^{-1}$, the individual molecules are recorded during about 50ms). However, at present, it's not clear whether this is an intrinsic gating property or a consequence of the crystal contacts affecting the gating energetics and stabilizing the two more stable conformations (active and resting). We think that by reconstituting the channel in crowded patches -rather than 2-D crystals- we should be able to address this issue and eventually capture less stable intermediates in the gating cycle (if they exist). However, this would require a substantial amount of work, ranging from sample preparation (finding the right lipids and reconstitution conditions to get the channels at the right density, avoiding crystals 2D-crystals formation, etc..) to data collection at much higher rates and analysis, implicating further developments. Intuitively, one might argue that the domain-swapped architecture of channel, and notably the fact that the C-linker forms an ensemble disk comprised of protein moiety from all 4 subunits, are indicative of a fully coordinated/simultaneous structural change. However, this question of the reviewer will only be answered satisfactorily with this technique if one indeed identifies intermediates in conditions of increased imaging rate and in non-crystalline packing. Absence of intermediates will not constitute evidence of concerted conformational changes since one can always argue that the intermediates are shorter than whatever the imaging rate is, and thus always shorter than what one can measure at the time. We detail some of these thoughts in the discussion of the revised version (page 14).

9. No AFM images were shown for SthK in the apo state. They should be shown. Is there any structural difference between the apo state and the cGMP-bound resting state?

Response: There are no obvious structural differences between the cGMP-bound state and apo state (compare Fig. 3h and Fig. S3b, dashed lines). Moreover, crystals imaged in the apo-state display the same lattice dimensions ($a=b=8.1\text{nm}$, $\gamma=90^\circ$) observed in the cGMP-bound state (see new panels in Figure S3). We consider these observations in the results section of the revised version (page 9 top).

10. Are the rate constants for on/off of cAMP and cGMP known? If known, the authors should discuss the observed structural changes in relation to these rate constants.

Response: We did not measure the on/off rate constants of the cyclic nucleotides for SthK and they are not reported in the literature. It has been previously reported for other channels such as MloK1 (Peuker et al., BJ 2013) and for the isolated CNBD of HCN2 (Goldschen-Ohm et al., eLife 2016), that the on-rate for cAMP binding is slower than expected for a diffusion limited rate, suggesting that it may be limited by a conformational change in the ligand binding pocket. It is entirely possible that, if this is true for SthK as well, then perhaps the reason why the conformational change from resting to activated states upon ligand application is so slow in these AFM experiments, is because the conformational change that limits the on-rate for cAMP/cGMP binding is restricted further by the crystal packing and thus the on-rate of cAMP is slowed even more than when the channels are free in solution. We consider this highly speculative however, given that we do not have values for these microscopic rate constants for SthK, so we did not think appropriate to speculate in the manuscript.

11. In Figure 3g, two different packing arrangements are seen within one 2D crystal patch. Considering the small area, the concentration of cGMP must be uniform over the entire area and all the molecules must have bound cGMP. Nevertheless, two distinct regions appear in one 2D crystal. What is the mechanism underlying this block separation between the two states?

Response: Thank you for this comment. The crystal shown in 3g was actually imaged in the presence of 0.1 mM cAMP. It was one of the handful of crystals displaying the two lattices simultaneously. We ascribe the underlying reason for the coexistence of the two forms to the inefficiency of cAMP to open the channel with high probability, and therefore the coexistence of the active and resting states even at saturating agonist concentrations and in a crystal where the selection for the activated conformation has not worked very well. The block separation probably arises from specific crystal contacts in the two lattices stabilizing one or the other conformation. We amended the Fig. 3 caption to clarify this.

Minor points

12. In Figure 4a, regions having very white (large height) objects appear. What are these objects?

Response: This is membrane bending, likely due to the molecules undergoing the conformational change (in $t=38\text{min}$, one can see that the channels are still there). We added text to indicate this in the figure legend of Fig. 4a.

13. Figure 1c shows the top view from the cytoplasmic side (Top) and the bottom view from the extracellular side (Bottom). However, in the side view the top is the extracellular side, while the bottom is the cytoplasmic side. These must be coincided with each other to avoid confusion.

Response: The reviewer is right, we interchanged the top and bottom views panels.

Reviewer 3

The present paper reports on an interesting set of experiments in which high speed AFM imaging is used to monitor the dynamics of global conformational changes in a cyclic-nucleotide gated channel. The authors show that the presence and absence of a ligand causes a concomitant rotational and upward movement of the cytosolic domain with respect to the part of the channel, which is harbored in the membrane. Even though this reviewer is not an expert of AFM measurements the data and the quantitative analysis appear convincing. In the field of ligand gated channels these are important data, because they finally provide some insights into long-range interactions and long-range conformational movements. This is important for an understanding of a communication between the ligand binding domain and the channel pore. For this reason I support publication of the manuscript.

Response: We thank the reviewer for this overall positive assessment of our work.

But before going into print the authors should first revise their manuscript with respect to a better presentation/discussion.

I strongly suggest the authors to restrict their interpretation to the gating of CNG channels. One of my main concerns of the paper is an over-interpretation of the data with respect to an invalid extrapolation to HCN channel gating. Other than stated in the 1st paragraph of the introduction, CN binding opens CNG channels but not HCN channels. HCN channels are opened by voltage and CN binding is only lowering the energetic barrier for activation. The gating mechanism of HCN channels cannot be discussed without information on the movement of the voltage sensor. The present data give a hint on the dynamics of the CNBD in HCN channels in response to ligand binding/dissociation. They do not explain gating of HCN channels.

Response: Having found a similar sequence similarity of SthK to CNG and HCN channels, we were tempted to extend our conclusions to both families. But the reviewer is correct: our results are less applicable to HCN channels. We remove these interpretations accordingly throughout the manuscript.

The authors should discuss their data in the context of previous studies, which already proposed a rotational movement of the CNBD \pm cAMP. Craven et al. (2008) Johnson & Zagotta 2001. The latter reference is cited in the text but in a different context.

Response: As recommended by the reviewer we have revised the discussion to include these previous works proposing a rotational movement of the CNBD/C-linker complex to gate the channel (page 12 bottom).

I do not understand the conditions of Fig. 2. The authors seem to observe the channel in the same preparation in two conformations. I don't see from the text if the protein was examined in the absence or presence of cAMP. Does the co-existence of both forms imply that the cAMP concentration (if present) is not saturating.

Response: There is a confusion here, and we modified the text (page 5) and the figure 2 legend to make it clearer. In figure 2, the two topographies show the same preparation viewed from the extracellular face (Fig. 2a-d) and the intracellular face (Fig. 2e-g), respectively, and not two different conformations. Both were imaged in the presence of 0.1 mM cAMP (as the figure title states), which is a saturating concentration (Schmidpeter et al., 2018). Both topographies display the same 2D crystal lattice geometries.

The information in Fig. 2 i and j is very important but difficult to see because of the colors in the images. Maybe the authors can improve the visibility of the rotational displacement between the proteins.

Response: We added a new panel in Fig. 2 to highlight the rotational displacement of the CNBDs.

REVIEWERS' COMMENTS:

Reviewer #1 (Remarks to the Author):

The authors have done a good job addressing my comments. I have no further concerns.

Reviewer #2 (Remarks to the Author):

The manuscript by Simon Scheuring and co-workers presents HS-AFM images of a cyclic nucleotide-gated ion channel (SthK) responding to the concentration changes of cAMP/cGMP. From the results, the authors have provided a mechanical gating model of this ion channel. My concerns and questions have been addressed satisfactorily by the addition of clearer explanations, which corroborates their initial conclusions. Thus, I think that the manuscript is now suitable for publication in Nature Communications.